# Personality profiles and meteoropathy intensity: A comparative study between young and older adults

**Marcin Rzeszutek**[ID]*, **Włodzimierz Oniszczenko**[ID], **Iwona Zalewska, Małgorzata Pięta**[ID]

Faculty of Psychology, University of Warsaw, Warsaw, Poland

* marcin.rzeszutek@psych.uw.edu.pl

## Abstract

### Objectives

This study's main aims were to investigate the Big Five personality trait heterogeneity of study participants in two age groups and to examine whether these traits' heterogeneity can explain possible individual differences in meteoropathy intensity.

### Method

The sample was comprised of 758 participants divided into two age groups: 378 young adults (18–30 years old) and 380 older adults (60+ years old). The participants filled out online or paper versions of the METEO-Q questionnaire and the Ten Item Personality Inventory (TIPI).

### Results

A latent profile analysis of the completed inventories showed various personality profiles differ in meteoropathy intensity. However, personality's differentiating effect on meteoropathy was observed only among the young adult group.

### Conclusions

Our study's results indicate that there is no one personality trait pattern that fits all individuals regarding sensitivity to weather changes. This issue is especially visible when considering age differences.

## Introduction

Starting with Hippocrates's theoretical observations in the book *Airs, Waters, and Places* [1], a large body of empirical evidence has shown the relationship between environmental factors, such as climate and weather changes, and human health and disease [see for review 2–4]. However, it was not until the turn of the 20[th] and 21[st] centuries when researchers began to

**Data Availability Statement:** The data are included in the Supporting Information.

**Funding:** This study was supported by the Faculty of Psychology, University of Warsaw, from the

funds awarded by the Ministry of Science and
Higher Education in the form of a subsidy for the
maintenance and development of research
potential in 2020 (501-D125-01-1250000.
zlec.5011000211/218).

**Competing interests:** The authors have declared
that no competing interests exist.

systematically study a new syndrome consisting of negative psychophysical symptoms related
to meteorological factors, that is, meteoropathy [5–7]. Specifically, meteoropathy refers to "a
group of symptoms and pathological reactions in response to gradual or sudden changes in
meteorological factors in a specific area interacting, presumably, through natural electromag-
netic influences covering a wide range of frequencies and amplitudes" [6, p. 46]. These reac-
tions, which last a few days and are strictly related to weather changes, include depressive
symptoms, irritability, numbness, sleep problems, muscles pain, and a general desire to remain
indoors [7]. Propensity to the aforementioned symptoms stems from meteorosensitivity, or
the body and mind's biological susceptibility to atmospheric changes [6]. Some studies have
observed that two demographic factors particularly contribute to individual differences in
meteorosensitivity, which are age [middle- and especially old-aged individuals; e.g., 8, 9] and
gender, with an overrepresentation of women in this regard [10, 11]. Not all meteorosensitive
people develop the pathological reactions that compose meteoropathy; nevertheless, until very
recently the mechanism responsible for meteoropathology at all was not entirely known [12].

In the last few years, some researchers found that superior vestibular nucleus activity may
be linked to meteoropathy in mice and probably in humans [13]. More concretely, they
claimed that the hypothalamus and amygdala nucleus can contribute to meteoropathy devel-
opment, meaning the human psychophysical propensity toward weather changes that induce
stress may be a derivative of individual differences in brain mechanisms responsible for emo-
tional regulation. In addition, it should be underscored that many signs of meteoropathy
reflect some mood disorders symptoms [7]. In line with this argument, it is vital to investigate
the role of personality as a potential correlate of meteoropathy [12], especially from the Big
Five taxonomy.

Dozens of studies have demonstrated the role of Big Five traits in emotional regulation [see
for review 14] and affective disorders in particular, which share similar biological dispositions
[15]. These studies highlighted the significance of neuroticism as predictor of maladaptive
emotional regulation and risk factor for mood disorders, as well as extraversion and conscien-
tiousness as buffering factors acting in the opposite way. However, Rammstedt et al. [16]
found that there can be just one Big Five trait that explains meteoropathy: openness to experi-
ence. Specifically, this trait strongly correlates with mood seasonality, and people attuned to
openness are among those who are most sensitive to perceived environmental changes, includ-
ing weather changes [16].

In our study, we investigated the role of the Big Five personality traits in the meteoropathy
intensity of two age groups: young and older adults. The methodological novelty of our study
featured the application of a person-centered perspective focused on the search for unique par-
ticipant profiles within the study variables. In the personality context, our person-centered
perspective enabled us to better understand the organization and functions of personality
within individuals' real lives [17]. This is most crucial, as all of the aforementioned meteoropa-
thy studies applied a variable-centered approach, which neglects the issue of participant het-
erogeneity regarding the examined variables.

## Current study

Taking the aforementioned research gaps into consideration, the aim of our study was twofold:
First, we investigated the heterogeneity of the study participants' Big Five personality traits.
Second, we assessed if these traits' heterogeneity can explain the possible individual differences
in meteoropathy intensity within the two participant age groups while controlling for their
sociodemographic data. To the best of our knowledge, there were no extant studies that may
have proven helpful as direct sources of research hypotheses in the case of this special study

design and particular outcome variable (i.e., meteoropathy). Therefore, this study mainly employed an exploratory approach. Nevertheless, based on extant literature conducted under the variable-centered approach, we expected our study samples to have heterogeneous Big Five personality traits and the observed participant profiles within these traits to declare different meteoropathy levels. Specifically, we hypothesized that the personality profiles of participants low in emotional stability and high in openness would on average have higher meteoropathy intensity, while the profiles of participants high in extraversion and high in conscientiousness would on average have lower meteoropathy intensity. We also assumed that older adults would have higher meteoropathy intensity compared to young adults. Additionally, we expected that female participants would suffer more from meteoropathy symptoms compared to male participants, regardless of age group.

## Method

### Participants and procedure

Our study sample was comprised of 758 participants divided into two nearly equinumerous age groups: 378 young adults (18–30 years old) and 380 older adults (60+ years old). The young adults were recruited from the general population via the online recruitment platform, where they filled out online versions of the study inventories (see the Measures section); there were no particular inclusion or exclusion criteria apart from falling under the young adult age bracket. The older adults were recruited by students from the various Universities of the Third Age in Warsaw, with their inventories gathered via lectures in paper format (see $n = 123$), as well as via the Facebook, where the participants had access to the online versions of our inventories ($n = 257$). Specifically, each of the University of the Third Age in Warsaw, where the study was conducted had its fan-page on Facebook, where we include the online link to our study. The older adults' inclusion criterion encompassed being at least 60 years old. The exclusion criterion included exhibited signs of dementia, which were screened for by clinical psychologists employed at the Universities of the Third Age, where this research was conducted via Mini-Mental State Examination (MMSE). Specifically, we choose those participants, who scored more at least 24 points in MMSE.

This study was anonymous, voluntary, and there was no renumeration for its participation. Informed consent was collected from all participants in the written form, which was included at the beginning of the study inventories both in the paper and pencil and in the online format of study questionnaires. The protocol of this study was accepted by the ethics committee of the Faculty of Psychology, University of Warsaw.

### Measures

Meteorosensitivity and meteoropathy were assessed via the Polish adaptation of the METEO-Q questionnaire [7]. This inventory consisted of 11 items that evaluate meteorosensitivity (5 items) and meteoropathy (6 items). The participants answered each item on a 4-point Likert scale ranging from 0 (*Absent*) to 3 (*Severe*). The Cronbach's alphas for the young and older adult samples are presented in Table 3.

The Big Five personality traits were examined with the Polish adaptation of the Ten Item Personality Inventory [TIPI; 18]. The TIPI measured each of the Big Five personality traits (extraversion, agreeableness, conscientiousness, emotional stability, and openness) in 2 items, with each assessed on a 7-point scale that ranged from 1 (*Disagree strongly*) to 7 (*Agree strongly*). The Cronbach's alphas for the young and older adult samples are presented in Table 3.

## Data analysis

Our data analysis consisted of three steps. First, the demographic characteristics of the analyzed sample, descriptive statistics, intercorrelations between analyzed variables, and differences between the young and older adult groups were computed. Deviation from normality was assessed with the skewness and kurtosis measures. In the next step, a latent profile analysis was performed to extract potential respondent subgroups and verify the first hypothesis [19]. The fit of models was assessed using the Aikake information criterion (AIC) and Bayesian information criterion (BIC). The acquired profiles were centered to foster clear interpretations. The final step involved examining the differences between the extracted classes in the young and older adult groups.

The descriptive statistics of the sample, analyzed variables, as well as the correlation analysis and differences between the young and older adult groups were performed using the IBM SPSS Statistics 26.0 software. The latent profile analysis was computed with the tidy LPA package, working in the R Statistics 4.0.1 environment.

## Results

Sociodemographic data of the two samples are presented in Table 1.

Table 2 presents the descriptive statistics (mean values, standard deviations, skewness and kurtosis measures, and intercorrelations between the analyzed variables) for the analyzed variables. None of the skewness or kurtosis measures exceeded the values of 1 or -1; we therefore assumed a normal analyzed variable distribution.

Table 3 presents the mean values of the analyzed variables in the young and older adult groups from Student's t-test.

**Table 1. Sociodemographic variables in the studied sample (N = 758).**

| | Young Adults | Older Adults | p |
|---|---|---|---|
| | (N = 378) | (N = 380) | |
| **Variable** | N (%) | N (%) | |
| Gender | | | |
| Male | 94 (24.8%) | 40 (10.5%) | .001 |
| Female | 284 (75.2%) | 340 (89.5%) | |
| Age in years (M ± SD) | 21.07 ± 2.31 | 67.58 ± 6.05 | .001 |
| Marital status | | | |
| Married | 14 (3.8%) | 135 (35.5%) | .001 |
| Single | 364 (96.2%) | 23 (6.1%) | |
| Informal relationship | 0 | 222 (58.4%) | |
| Education | | | |
| Elementary | 83 (21.9%) | 13(3.5%) | .001 |
| Secondary | 281 (74.3%) | 189 (49.7%) | |
| Higher education | 14 (3.8%) | 178 (46.8%) | |
| Place of residence | | | |
| Village, small town up to 20,000 residents | 93 (24.6%) | 64 (16.8%) | .010 |
| City, 21,000–100,000 residents | 64 (17.0%) | 60 (15.8%) | |
| City, 101,000–500,000 residents | 34 (9.0%) | 56 (14.7%) | |
| City, over 500,000 residents | 187 (49.4%) | 200 (52.6%) | |

*Note*: *M* = mean; *SD* = standard deviation; *p* = p-value of chi-squared test for independence or Student's independent sample t-test of participant age.

**Table 2. Descriptive statistics and pearson correlation coefficients between analyzed variables in the full participant sample (N = 758).**

| Variables | M | SD | S | K | 1. | 2. | 3. | 4. | 5. | 6. | 7. |
|---|---|---|---|---|---|---|---|---|---|---|---|
| 1. Emotional stability | 7.82 | 3.45 | 0.08 | -0.92 | - | | | | | | |
| 2. Extroversion | 10.02 | 3.27 | -0.61 | -0.64 | .471** | - | | | | | |
| 3. Openness to experience | 9.31 | 2.51 | -0.35 | 0.20 | .105** | .322** | - | | | | |
| 4. Agreeableness | 10.82 | 2.55 | -0.78 | 0.18 | .245** | .283** | .107** | - | | | |
| 5. Conscientiousness | 10.40 | 3.09 | -0.73 | -0.38 | .238** | .380** | .065 | .302** | - | | |
| 6. Meteorosensitivity | 11.21 | 4.48 | -0.21 | -0.34 | .060 | .213** | -.019 | .153** | .224** | - | |
| 7. Meteoropathy | 9.85 | 4.43 | -0.15 | -0.50 | .043 | .207** | -.048 | .144** | .263** | .844** | - |

*Note*: *M* = mean value; *SD* = standard deviation; *S* = skewness; *K* = kurtosis; * $p < .05$

** $p < .01$.

According to the Student's t-test values for the independent samples, there was a statistical difference between men and women regarding meteorosensitivity ($t(754) = 7.94$, $p < .001$) and meteoropathy ($t(754) = 9.22$, $p < .001$). For women, meteorosensitivity ($M = 11.78$, $SD = 4.26$) and meteoropathy ($M = 10.50$, $SD = 4.16$) were significantly higher than for men, for whom the mean meteorosensitivity value was 8.51 ($SD = 4.52$) and that of meteoropathy 6.78 ($SD = 4.43$).

Next, we executed a latent profile analysis to estimate distinct personality profiles and extract subgroups of respondents differing in meteorosensitivity and meteoropathy. The analysis was performed separately for the young and older adult groups. According to an analytic hierarchy process and based on the fit indices AIC, AWE, BIC, CLC, and KIC [20], the best solution for the young adult group was a 3-class model with differing variances and covariances. The fit statistics values were AIC = 5,050.55 and BIC = 5,294.19. Fig 1 presents the mean values of the standardized variables in the three acquired classes.

In the first class, the acquired profile was characterized by average levels of all personality traits (*profile 1*). In the second class, the acquired profile was characterized by a high level of conscientiousness (*profile* 2). In the third class, the acquired profile was characterized by low emotional stability (*profile* 3).

In the older adult group, according to an analytic hierarchy process and based on the fit indices AIC, AWE, BIC, CLC, and KIC, the best solution was a 2-class model with differing variances and covariances. The fit statistics values were AIC = 4,834.64 and BIC = 4,996.19. Fig 2 presents the mean values of the standardized variables in the acquired classes.

**Table 3. Values of analyzed variables in the young and older adult groups with the values of student's independent sample t-test.**

| Variables | Young Adults | | | Older Adults | | | t | df | p | d |
|---|---|---|---|---|---|---|---|---|---|---|
| | M | SD | α | M | SD | α | | | | |
| Emotional stability | 6.38 | 3.31 | .75 | 9.23 | 2.96 | .62 | -12.48 | 743.34 | .001 | -.91 |
| Extroversion | 8.77 | 3.26 | .75 | 11.25 | 2.77 | .69 | -11.29 | 732.45 | .001 | -.82 |
| Openness to experience | 9.30 | 2.47 | .74 | 9.32 | 2.55 | .62 | -0.11 | 754 | .910 | -.01 |
| Agreeableness | 10.30 | 2.58 | .60 | 11.34 | 2.42 | .61 | -5.72 | 754 | .001 | -.42 |
| Conscientiousness | 9.30 | 3.18 | .76 | 11.48 | 2.59 | .61 | -10.35 | 721.25 | .001 | -.75 |
| Meteorosensitivity | 8.46 | 3.53 | .82 | 13.93 | 3.56 | .83 | -21.23 | 754 | .001 | -1.54 |
| Meteoropathy | 7.29 | 3.81 | .76 | 12.37 | 3.45 | .79 | -19.20 | 744.98 | .001 | -1.40 |

*Note*: *M* = mean value; *SD* = standard deviation; *t* = independent sample Student's t-test; α = Cronbach's alpha; *df* = degrees of freedom; *p* = statistical significance; *d* = Cohen's effect size measure.

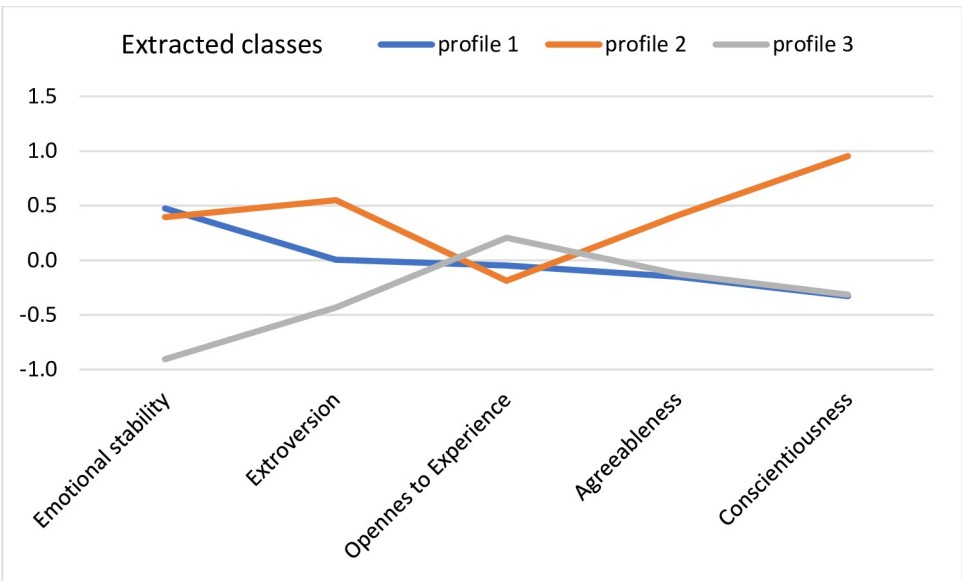

**Fig 1. Profiles of personality acquired in the group of young adults.**

In the first class, the acquired profile was characterized by low levels of extroversion and conscientiousness (*profile 1*). In the second class, the acquired profile was characterized by high levels of extroversion and conscientiousness (*profile 2*).

The three extracted young adult classes were compared in terms of meteorosensitivity and meteoropathy. Taking into account the participants' genders, an analysis of covariance was performed. Between groups, differences were statistically significant for both meteorosensitivity ($F(2,372) = 7.68$, $p < .01$, $\eta^2 = .04$) and meteoropathy ($F(2,372) = 3.54$, $p < .05$, $\eta^2 = .02$). According to the values of a multiple comparison test with Bonferroni adjustment, the third

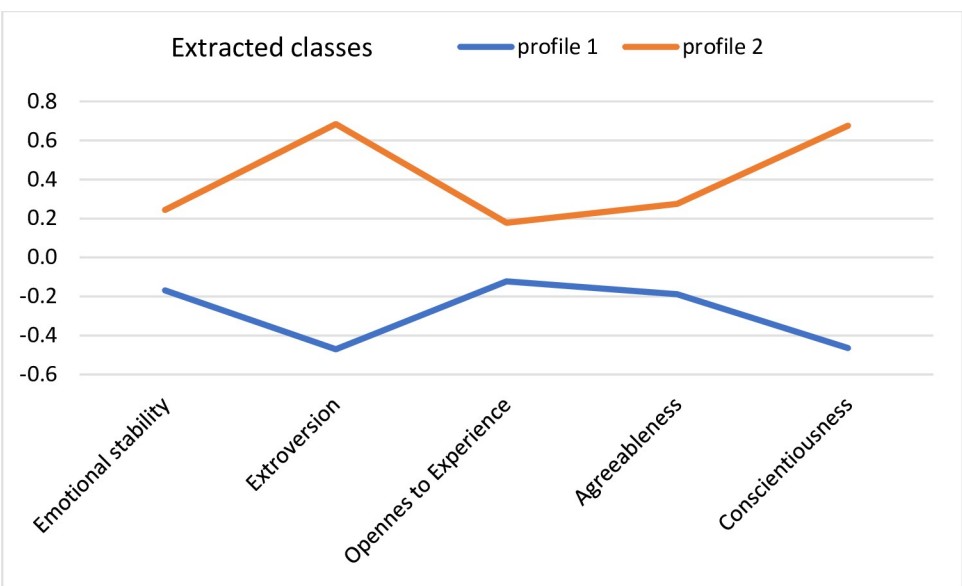

**Fig 2. Profiles of personality acquired in the group of older adults.**

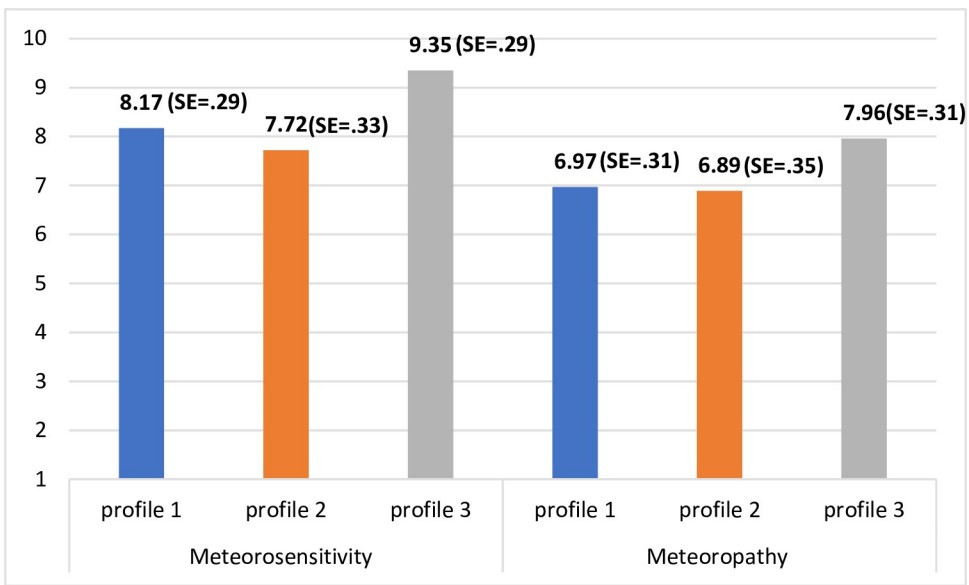

**Fig 3. Mean values of meteorosensitivity and meteoropathy.** Acquired in the extracted classes in the group of young adults.

class (*profile 3*) significantly differed from the first ($p < .01$) and the second ($p < .05$). The mean meteorosensitivity and meteoropathy values were significantly higher in the third class, with the participants having lower levels of emotional stability (see Fig 3).

In the older adult group, the differences between extracted classes were statistically insignificant regarding both meteorosensitivity ($F(1,377) = 1.33$, $p > .01$) and meteoropathy ($F(1,377) = .10$, $p > .05$).

## Discussion

Our study's results were consistent with our first hypothesis to a point, as, on the one hand, we managed to observe Big Five personality trait heterogeneity across our two study samples. On the other hand, though, the extracted personality profiles differed with regard to meteoropathy intensity only within the young adult sample. Specifically, in this age group, we found three personality profiles: participants with average levels of all five personality traits (*profile 1*), participants with very high conscientiousness levels and average levels of other traits (*profile 2*), and participants with low emotional stability and average levels of other traits (*profile 3*). Interestingly, we observed significant differences between these profiles in both meteorosensitivity and meteoropathy. Namely, young adults from *profile 3* declared much higher meteorosensitivity levels than those from the other two profiles, as well as much higher meteoropathy levels than those from *profile 1*. It seems that contrary to our hypothesis, among young people, only one personality trait matters with regard to meteoropathy: neuroticism. This finding is not as obvious as it seems at first glance, as while research on weather sensitivity, particularly on seasonal affective disorders, has indeed highlighted neuroticism's strong predictive role [21, 22], especially how this trait is characterized by the experience of strong mood variability [23], more recent daily diary studies have identified that personality traits cannot account for weather sensitivity [24]. According to these latter authors, weather sensitivity is somehow an *individual difference factor by itself*, as people can have different weather sensitivity thresholds regardless of their personality traits. Furthermore, these authors claimed the same of

sociodemographic data, including especially age and gender. Still, each of these studies applied a variable-centered approach, so they assumed a priori their study samples' homogeneity. We believe that following a person-centered perspective can provide a more realistic look at the interrelationships between variables, which was the focus of our research.

The picture of the association between personality and meteoropathy in the older adult group was very different compared to the young adult group. We observed two personality profiles among the older adults: participants with low extraversion and conscientiousness intensity and average levels of other traits (*profile 1*) and participants with high extraversion and conscientiousness intensity and average levels of other traits (*profile 2*). Among the older adults, however, the extracted personality profiles did not differ in meteorosensitivity or meteoropathy. Still, it should be noted that the older adults declared much higher meteorosensitivity and meteoropathy severity in comparison to the young adults, which was in line with our second hypothesis and other studies in this area [8, 9, 12, 25]. The effect sizes of these differences were quite large (see Table 3), so we carefully assume that while personality profiles differed in both meteorosensitivity and meteoropathy levels in the young adult group, among older adults, meteorosensitivity and meteoropathy severity is high, and the seniors themselves vary so little with regard to these constructs that personality traits no longer matter.

Lastly, we found higher meteorosensitivity and meteoropathy intensity among the female participants in both age groups. This result is not only consistent with our final hypothesis, but also reflects other authors' observations [11, 26] for example, Connolly [10] showed that women are more prone to reacting to environmental factors, which relates especially to poor life satisfaction. Interestingly, low life satisfaction from several domains of functioning (e.g., job and health situation) is the worst on rainy days. Despite our results, the gender and non-clinical sample mechanisms in this regard are still not entirely known and require further examination.

## Strengths and limitations

This study has several strengths, including its innovative methodology and hypothesis-driven design utilizing several understudied constructs in a large sample of two age groups. However, it is vital to pinpoint some of our research's limitations. First, in the older adult group, we did not thoroughly screen our participants' actual health statuses, although we excluded patients exhibiting signs of dementia. Second, in the young adult group, we did not control for potential mental disorders or substance abuse. Thirdly, our study samples significantly differed across analyzed socio-demographic characteristics. Furthermore, not every participant underwent the psychometric assessment through the same modality (paper and pencil format versus online version), which could raise some potential bias in the obtained results. One final limitation is this study's cross-sectional and self-report measures.

## Conclusion

Even with these limitations, our study's results may shed new perspective on the link between personality and the still much understudied syndrome meteoropathy. In light of our findings, it seems there is no one personality trait pattern that fits all individuals in the context of sensitivity to weather changes. This latter issue is especially visible when we take age differences into account.

## Supporting information

**S1 Dataset.**
(SAV)

## Author Contributions

**Conceptualization:** Marcin Rzeszutek, Włodzimierz Oniszczenko, Iwona Zalewska.

**Data curation:** Marcin Rzeszutek, Iwona Zalewska, Małgorzata Pięta.

**Formal analysis:** Marcin Rzeszutek.

**Methodology:** Włodzimierz Oniszczenko.

**Project administration:** Marcin Rzeszutek, Iwona Zalewska, Małgorzata Pięta.

**Resources:** Małgorzata Pięta.

**Supervision:** Marcin Rzeszutek, Iwona Zalewska.

**Visualization:** Małgorzata Pięta.

**Writing – original draft:** Marcin Rzeszutek, Włodzimierz Oniszczenko.

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
