## [Decision Letter · Decision Letter 0]

3 Sep 2020

PONE-D-20-20771

Personality Profiles and Meteoropathy Intensity: A Comparative Study between Young and Older Adults

PLOS ONE

Dear Dr. Rzeszutek,

Thank you for submitting your manuscript to PLOS ONE. After careful consideration, we feel that it has merit but does not fully meet PLOS ONE’s publication criteria as it currently stands. Therefore, we invite you to submit a revised version of the manuscript that addresses the points raised during the review process.

We look forward to receiving your revised manuscript.

Kind regards,

Geilson Lima Santana, M.D., Ph.D.

Academic Editor

PLOS ONE

Journal Requirements:

2. Please provide additional details regarding participant consent. In the ethics statement in the Methods and online submission information, please ensure that you have specified what type you obtained (for instance, written or verbal, and if verbal, how it was documented and witnessed). If the need for informed written consent was waived by the ethics committee, please include this information.

Reviewers' comments:

Reviewer's Responses to Questions

**Comments to the Author**

1. Is the manuscript technically sound, and do the data support the conclusions?

Reviewer #1: Yes

Reviewer #2: Partly

2. Has the statistical analysis been performed appropriately and rigorously? 

Reviewer #1: Yes

Reviewer #2: Yes

3. Have the authors made all data underlying the findings in their manuscript fully available?

Reviewer #1: Yes

Reviewer #2: Yes

4. Is the manuscript presented in an intelligible fashion and written in standard English?

Reviewer #1: Yes

Reviewer #2: No

5. Review Comments to the Author

Reviewer #1: This is a very intersting study investigating personality trait heterogeneity

in two age groups (young and older adults) and examining whether traits’ heterogeneity can

explain possible individual differences in meteoropathy intensity.

The research is well conducted and clear.

a minor comment:

page 5 section Current Study "we expected, we expected..." please correct

Reviewer #2: Dear Authors,

meteorosensitivity and meteoropathy have progressively become an issue of increasing interest, with the goal of better understanding the underlining biological and psychopathological features and developing targeted intervention strategies in the feature.

First, the covered topic is of course timely, and the sample size is adequate. As well, the main strength of the paper is the special study design focusing on personality profiles, with interesting results. However, some point-to-point issues must be acknowledged to carry out your revision.

Methods:

-Middle-aged and, above all, old-aged individuals are at greater risk of meteorosensitivity/meteoropathy, as mentioned in the Introduction. Why did you chose to analyze young adults (18-30 years old) in your study, as well as to exclude adults ranging from 30 to 60 years old? Please justify your choice in the response to reviewers (rather than in the paper).

-Have the METEO-Q and TIPI instruments been validated in Polish? If not, this is a potential source of flaw and it should be included in the study limitations.

-Not every participant underwent the psychometric assessment through the same modality (paper format vs. online version), rising further potential bias that should be mentioned in the limitations paragraph.

-Study recruitment in the older adults group is not entirely clear. Please, clarify methods about the use of Facebook to get the psychometric assessment and concomitant screening of dementia signs by the University psychologists.

-The two groups significantly differed in all their socio-demographic characteristics, with no express mention in the text. Moreover, Table 1 about sociodemographic data of the two samples should be moved to the Results section.

Discussion:

As pointed out in the manuscript, to date there is a lack of sufficient data on both personality traits and meteorosensitivity/meteoropathy to formulate a clear hypothesis about the common biological mechanisms of these phenomena. However, we would like to recommend the following studies in order to deepen the strength of your discussion:

- Di Nicola M, Mazza M, Panaccione I, Moccia L, Giuseppin G, Marano G, Grandinetti P, Camardese G, De Berardis D, Pompili M, Janiri L. Sensitivity to Climate and Weather Changes in Euthymic Bipolar Subjects: Association With Suicide Attempts. Front Psychiatry. 2020 Mar 5;11:95. doi: 10.3389/fpsyt.2020.00095. PMID: 32194448; PMCID: PMC7066072;

- Cianconi P, Betrò S, Grillo F, Hanife B, Janiri L. Climate shift and mental health adjustment [published online ahead of print, 2020 Apr 6]. CNS Spectr. 2020;1-2. doi:10.1017/S1092852920001261.

In general, the authors should tone down the speculation on the hypotheses by which certain personality traits may relate or not to meteorosensitivity/meteoropathy (e.g., “we agree with Denissen … but related only to the older adult group”).

Some inconsistencies between text and reference list should be solved too (e.g., Gossling et al., 2003, Akogul and Erisoglu, 2017; they are cited in the manuscript, but they are not reported in the References section). We also recommend checking out all the other ones.

To conclude, an extensive revision by a native speaker with technical-scientific linguistic skills is desirable.

Best regards.

6. PLOS authors have the option to publish the peer review history of their article (what does this mean?). If published, this will include your full peer review and any attached files.

Reviewer #1: **Yes: **Marianna Mazza

Reviewer #2: No

---

## [Author Response · Author response to Decision Letter 0]

5 Sep 2020

Dear Editor, Dear Reviewers, 

thank you very much for suggestions and remarks concerning our article titled “Personality Profiles and Meteoropathy Intensity: A Comparative Study between Young and Older Adults”, which we would like to publish in PLOS One. We referred to all reviewers’ remarks. Below we cite every remark and comment of the reviewers and provide the answers to them in parentheses. All the changes in the revised text are marked with red font.

Editor remarks

[Thank you very much for reminder. We double checked that our manuscript meets PLOS ONE's style requirements.]

2. Please provide additional details regarding participant consent. In the ethics statement in the Methods and online submission information, please ensure that you have specified what type you obtained (for instance, written or verbal, and if verbal, how it was documented and witnessed). If the need for informed written consent was waived by the ethics committee, please include this information.

[Thank you very much for this remark. In the revised version of the manuscript we provided more details on participant consent. More specifically, informed consent was collected from all participants in the written form, which was included at the beginning of the study inventories both in the paper and pencil and online format of study questionnaires. The ethics committee did not underline the particular form of informed consent (i.e. verbal or written), but only underscore the necessity for obtaining it either in written or verbal mode.]

Comments to the Author

1. Is the manuscript technically sound, and do the data support the conclusions?

Reviewer #1: Yes

Reviewer #2: Partly

2. Has the statistical analysis been performed appropriately and rigorously?

Reviewer #1: Yes

Reviewer #2: Yes

3. Have the authors made all data underlying the findings in their manuscript fully available?

Reviewer #1: Yes

Reviewer #2: Yes

4. Is the manuscript presented in an intelligible fashion and written in standard English?

Reviewer #1: Yes

Reviewer #2: No

[Thank you very much for these positive, general, evaluation of our manuscript. As far as the remark of the Reviewer 2 on not sufficiently professional level of English, we would like to underline the fact that before sending our manuscript to PLOS ONE, we sent our manuscript to the professional English proofreading service. However, regarding your criticism, once again we sent the revised version of our manuscript to another Native English speaker, who corrected some additional language errors. Thus, please bear in mind that we did our utmost to prepare our manuscript in the best English version as we can.]

Reviewer #1: 

This is a very interesting study investigating personality trait heterogeneity

in two age groups (young and older adults) and examining whether traits’ heterogeneity can

explain possible individual differences in meteoropathy intensity.

The research is well conducted and clear. A minor comment:

page 5 section Current Study "we expected, we expected..." please correct

[Thank you very much for so positive feedback on our manuscript. We really appreciated it. Regarding this minor remark, we corrected this small typo in the current study – thanks for paying attention on this.]

Reviewer #2: 

Dear Authors,

meteorosensitivity and meteoropathy have progressively become an issue of increasing interest, with the goal of better understanding the underlining biological and psychopathological features and developing targeted intervention strategies in the feature.

First, the covered topic is of course timely, and the sample size is adequate. As well, the main strength of the paper is the special study design focusing on personality profiles, with interesting results. However, some point-to-point issues must be acknowledged to carry out your revision.

[Thank you very much for generally positive feedback on our manuscript.]

Methods:-Middle-aged and, above all, old-aged individuals are at greater risk of meteorosensitivity/meteoropathy, as mentioned in the Introduction. Why did you chose to analyze young adults (18-30 years old) in your study, as well as to exclude adults ranging from 30 to 60 years old? Please justify your choice in the response to reviewers (rather than in the paper).

[Thank you very much for your question. There was one particular reason explaining why we choose this two, contrasting age group were two-fold - it was about the comparison of extreme groups or more differentiated in terms of age according to the paradigm of individual differences and the studies on that syndrome. The group of young and old 60+ should be supposed to be the most different in terms of meteorosensitivity and, of course, meteoropathy.]

-Have the METEO-Q and TIPI instruments been validated in Polish? If not, this is a potential source of flaw and it should be included in the study limitations.

[Yes, the METEO-Q was validated in Poland by Oniszczenko – in one unpublishe paper from 2019, as well in published paper below:

Oniszczenko, W. (2020), Affective temperaments and meteoropathy among women: A cross-sectional study. PLoS ONE, 15, e0232725. doi.org/10.1371/journal. pone.0232725

And the TIPI by: Sorokowska, A., Słowińska A., Zbieg A., Sorokowski, P. (2014). Polska adaptacja testu Ten Item Personality Inventory (TIPI) – TIPI-PL – wersja standardowa i internetowa. Wrocław: WrocLab (page 25). [in Polish]

However, as we stated in the cover letter, details information regarding Polish adaptations of questionnaires used in the study were not included in the submitted version of the manuscript in order to support a truly blind reviewing process. However, in the final version of the manuscript in PLOS ONE we can reveal all these details.]

-Not every participant underwent the psychometric assessment through the same modality (paper format vs. online version), rising further potential bias that should be mentioned in the limitations paragraph.

[It is very important remark. Due to our wish to get access to the wide variety of study participants, which was especially related of older adults (which are not always “online-familiar”), we combined the paper-and pencil format with online modality. However, we agree that it could potentially lead to bias, which we mentioned in the limitations section.]

-Study recruitment in the older adults group is not entirely clear. Please, clarify methods about the use of Facebook to get the psychometric assessment and concomitant screening of dementia signs by the University psychologists.

[Thank you very much for this question. In the revised version of the manuscript we specified that the older adults were recruited by students from the various Universities of the Third Age in Warsaw, with their inventories gathered via lectures in paper format (see n = 123), as well as via the Facebook, where the participants had access to the online versions of our inventories (n = 257). Specifically, each of the University of the Third Age in Warsaw, where the study was conducted had its fan-page on Facebook, where we include the online link to our study. As far as the possible signs of dementia, they were screened for by clinical psychologists employed at the Universities of the Third Age, where this research was conducted via Mini-Mental State Examination.]

-The two groups significantly differed in all their socio-demographic characteristics, with no express mention in the text. Moreover, Table 1 about sociodemographic data of the two samples should be moved to the Results section.

[It is very important remark. Due to the significant age differences, the socio-demographic data cannot be the same or even similar across these two samples. However, we mentioned about thus fact in the study limitations. We also moved the Table 1to the Results section, in accordance to your suggestion.]

Discussion:

As pointed out in the manuscript, to date there is a lack of sufficient data on both personality traits and meteorosensitivity/meteoropathy to formulate a clear hypothesis about the common biological mechanisms of these phenomena. However, we would like to recommend the following studies in order to deepen the strength of your discussion:

- Di Nicola M, Mazza M, Panaccione I, Moccia L, Giuseppin G, Marano G, Grandinetti P, Camardese G, De Berardis D, Pompili M, Janiri L. Sensitivity to Climate and Weather Changes in Euthymic Bipolar Subjects: Association With Suicide Attempts. Front Psychiatry. 2020 Mar 5;11:95. doi: 10.3389/fpsyt.2020.00095. PMID: 32194448; PMCID: PMC7066072;

- Cianconi P, Betrò S, Grillo F, Hanife B, Janiri L. Climate shift and mental health adjustment [published online ahead of print, 2020 Apr 6]. CNS Spectr. 2020;1-2. doi:10.1017/S1092852920001261.

[Thank you for this literature, We included aforementioned studies in the revised version of manuscript.]

In general, the authors should tone down the speculation on the hypotheses by which certain personality traits may relate or not to meteorosensitivity/meteoropathy (e.g., “we agree with Denissen … but related only to the older adult group”).

[Yes – you were right to tone down with this speculations – we improved this part to avoid speculative remarks.]

Some inconsistencies between text and reference list should be solved too (e.g., Gossling et al., 2003, Akogul and Erisoglu, 2017; they are cited in the manuscript, but they are not reported in the References section). We also recommend checking out all the other ones.

To conclude, an extensive revision by a native speaker with technical-scientific linguistic skills is desirable.

[Thank you for paying our attention on these inconsistencies -we improved this errors in references. As far as the remark of on the language, we would like to underline again the fact that before sending our manuscript to PLOS ONE, we sent our manuscript to the professional English proofreading service. However, regarding your criticism, once again we sent the revised version of our manuscript to another Native English speaker, who corrected some additional language errors. Thus, please bear in mind that we did our utmost to prepare our manuscript in the best English version as we can.]

To sum up, I would like to thank Editor and Reviewers for their time and effort. We found all the comments very useful and I believe that they helped me and my co-authors to improve the manuscript quality. I deeply appreciate a chance you gave us to revise and submit it to be considered for publication in PLOS One.

---

## [Decision Letter · Decision Letter 1]

30 Sep 2020

PONE-D-20-20771R1

Personality Profiles and Meteoropathy Intensity: A Comparative Study between Young and Older Adults

PLOS ONE

Dear Dr. Rzeszutek,

Thank you for submitting your manuscript to PLOS ONE. After careful consideration, we feel that it has merit but does not fully meet PLOS ONE’s publication criteria as it currently stands. Therefore, we invite you to submit a revised version of the manuscript that addresses the points raised during the review process.

We look forward to receiving your revised manuscript.

Kind regards,

Geilson Lima Santana, M.D., Ph.D.

Academic Editor

PLOS ONE

Reviewers' comments:

Reviewer's Responses to Questions

**Comments to the Author**

1. If the authors have adequately addressed your comments raised in a previous round of review and you feel that this manuscript is now acceptable for publication, you may indicate that here to bypass the “Comments to the Author” section, enter your conflict of interest statement in the “Confidential to Editor” section, and submit your "Accept" recommendation.

Reviewer #1: All comments have been addressed

Reviewer #2: (No Response)

2. Is the manuscript technically sound, and do the data support the conclusions?

Reviewer #1: Yes

Reviewer #2: Yes

3. Has the statistical analysis been performed appropriately and rigorously? 

Reviewer #1: Yes

Reviewer #2: Yes

4. Have the authors made all data underlying the findings in their manuscript fully available?

Reviewer #1: Yes

Reviewer #2: Yes

5. Is the manuscript presented in an intelligible fashion and written in standard English?

Reviewer #1: Yes

Reviewer #2: Yes

6. Review Comments to the Author

Reviewer #1: (No Response)

Reviewer #2: Dear Authors, we have read and appreciated your manuscript’s improvements. Finally, further efforts are required as follows:

- To be clearer, the phrase at lines 83-85 could be adjusted in this way: “These studies highlighted the significance of neuroticism as predictor of maladaptive emotional regulation and risk factor for mood disorders, as well as extraversion and conscientiousness as buffering factors acting in the opposite way.”

- At line 132, please specify the MMSE cut-off score which was used to detect/exclude patients with dementia (e.g., MMSE <26)

- Please, “The sociodemographic data of the two samples are presented in Table 1. [Insert Table 1 about here]” at lines 137-139 should be moved to Results section, as previously requested.

- Di Nicola et al., 2020 (“Sensitivity to Climate and Weather Changes in Euthymic Bipolar Subjects: Association With Suicide Attempts.”) should be moved to lines 254-258, in relation to meteorosensitivity/meteoropathy intensity among female subjects.

- Please, be sure to include references about Polish validations of the psychometric instruments (METEO-Q and TIPI) in the final version of your manuscript.

7. PLOS authors have the option to publish the peer review history of their article (what does this mean?). If published, this will include your full peer review and any attached files.

Reviewer #1: **Yes: **Marianna Mazza

Reviewer #2: No

---

## [Author Response · Author response to Decision Letter 1]

1 Oct 2020

Dear Editor, Dear Reviewers, 

thank you very much for another suggestions and remarks concerning our article titled “Personality Profiles and Meteoropathy Intensity: A Comparative Study between Young and Older Adults”, which we would like to publish in PLOS One. We referred to all reviewers’ remarks. Below we cite every remark and comment of the reviewers and provide the answers to them in parentheses. All the changes in the revised text are marked with red font.

Reviewer #2: Dear Authors, we have read and appreciated your manuscript’s improvements. 

[Thank you for kind words.]

Finally, further efforts are required as follows:

- To be clearer, the phrase at lines 83-85 could be adjusted in this way: “These studies highlighted the significance of neuroticism as predictor of maladaptive emotional regulation and risk factor for mood disorders, as well as extraversion and conscientiousness as buffering factors acting in the opposite way.”

[We changed this sentence according to your wish.]

- At line 132, please specify the MMSE cut-off score which was used to detect/exclude patients with dementia (e.g., MMSE <26).

[We included the cut-off score for our participants.]

- Please, “The sociodemographic data of the two samples are presented in Table 1. [Insert Table 1 about here]” at lines 137-139 should be moved to Results section, as previously requested.

[Thank you for this remark – we moved the Table 1 to the Results section.]

- Di Nicola et al., 2020 (“Sensitivity to Climate and Weather Changes in Euthymic Bipolar Subjects: Association With Suicide Attempts.”) should be moved to lines 254-258, in relation to meteorosensitivity/meteoropathy intensity among female subjects.

[Thank you for this remark – we moved this paper in relation to meteorosensitivity/meteoropathy intensity among female subjects.]

- Please, be sure to include references about Polish validations of the psychometric instruments (METEO-Q and TIPI) in the final version of your manuscript.

[Of course – in the final version of the article we will include the Polish validations of the psychometric instruments.]

To sum up, I would like to thank Editor and Reviewers for their time and effort. We found all the comments very useful and I believe that they helped me and my co-authors to improve the manuscript quality. I deeply appreciate a chance you gave us to revise and submit it to be considered for publication in PLOS One.

---

## [Decision Letter · Decision Letter 2]

21 Oct 2020

Personality Profiles and Meteoropathy Intensity: A Comparative Study between Young and Older Adults

PONE-D-20-20771R2

Dear Dr. Rzeszutek,

We’re pleased to inform you that your manuscript has been judged scientifically suitable for publication and will be formally accepted for publication once it meets all outstanding technical requirements.

Kind regards,

Geilson Lima Santana, M.D., Ph.D.

Academic Editor

PLOS ONE

Additional Editor Comments (optional):

Reviewers' comments:

Reviewer's Responses to Questions

**Comments to the Author**

1. If the authors have adequately addressed your comments raised in a previous round of review and you feel that this manuscript is now acceptable for publication, you may indicate that here to bypass the “Comments to the Author” section, enter your conflict of interest statement in the “Confidential to Editor” section, and submit your "Accept" recommendation.

Reviewer #1: All comments have been addressed

Reviewer #2: (No Response)

2. Is the manuscript technically sound, and do the data support the conclusions?

Reviewer #1: Yes

Reviewer #2: (No Response)

3. Has the statistical analysis been performed appropriately and rigorously? 

Reviewer #1: Yes

Reviewer #2: (No Response)

4. Have the authors made all data underlying the findings in their manuscript fully available?

Reviewer #1: Yes

Reviewer #2: (No Response)

5. Is the manuscript presented in an intelligible fashion and written in standard English?

Reviewer #1: Yes

Reviewer #2: (No Response)

6. Review Comments to the Author

Reviewer #1: (No Response)

Reviewer #2: (No Response)

7. PLOS authors have the option to publish the peer review history of their article (what does this mean?). If published, this will include your full peer review and any attached files.

Reviewer #1: **Yes: **Marianna Mazza

Reviewer #2: No

---

## [Editor Report · Acceptance letter]

10 Nov 2020

PONE-D-20-20771R2 

Personality Profiles and Meteoropathy Intensity:A Comparative Study between Young and Older Adults 

Dear Dr. Rzeszutek:

I'm pleased to inform you that your manuscript has been deemed suitable for publication in PLOS ONE. Congratulations! Your manuscript is now with our production department. 

Kind regards, 

on behalf of

Dr. Geilson Lima Santana 

Academic Editor

PLOS ONE